# Effectiveness and Efficacy of Long-Lasting Insecticidal Nets for Malaria Control in Africa: Systematic Review and Meta-Analysis of Randomized Controlled Trials

**DOI:** 10.3390/ijerph22071045

**Published:** 2025-06-30

**Authors:** Dereje Bayisa Demissie, Getahun Fetensa, Tilahun Desta, Firew Tiruneh Tiyare

**Affiliations:** 1Department of Nursing and Midwifery, Faculty of Medicine and Health Sciences, Stellenbosch University, Cape Town 7602, South Africa; 2Department of Medical Surgical Nursing, School of Nursing, Saint Paul’s Hospital Millennium Medical College, Addis Ababa P.O. Box 1271, Ethiopiatilahundesta84@yahoo.com (T.D.); 3Department of Epidemiology, Institute of Health, Jimma University, Jimma P.O. Box 5195, Ethiopia; mtu2012x@gmail.com

**Keywords:** effectiveness, efficacy, long-lasting insecticidal nets, pyriproxyfen, chlorfenapyr, piperonyl butoxide, pyrethroid only, malaria control, Africa

## Abstract

**Background**: Long-lasting insecticidal nets (LLINs) have significantly reduced the malaria burden in recent decades, and this malaria prevalence reduction has been achieved through the upgrading of pyrethroid long-lasting insecticidal nets. However, this reduction has stalled due to many factors, including rapidly developing pyrethroid resistance. **Method:** The protocol was registered in PROSPERO, and we used Cochrane methodology to assess bias and evidence quality. Three reviewers extracted data from individual studies, and a meta-analysis was performed using Excel and STATA version 17, expressing the data as a risk ratio. **Result**: A study involving 21,916 households from 11 randomized controlled trials showed that the chlorfenapyr treatment group had a 10% reduction in malaria infection risk, with a pooled overall prevalence of 25.96 per 100 children in the chlorfenapyr group and 32.38 per 100 children in the piperonyl butoxide group, compared to 41.60 per 100 children in the control (pyrethroid-only) group. This meta-analysis determined that the entomological outcomes of effectiveness and efficacy showed that these treatments effectively reduced vector density per household per night and mean inoculation rates, with a 23% reduction in chlorfenapyr, a 7% reduction in pyrethroid-only treatments, and a 12% reduction in piperonyl-butoxide-treated groups. This study shows that chlorfenapyr (CFP) and pyriproxyfen (PPF) LLINs are highly effective and more efficacious in reducing malaria infection, case incidence, and anemia among children, as well as in reducing mean indoor vector density, mean entomological inoculation rate, and sporozoite rate, compared to pyriproxyfen (PPF) LLINs in Africa. **Conclusions**: This study found that chlorfenapyr (CFP) LLINs are highly effective and more efficacious in reducing malaria infection, case incidence, and anemia among children in Africa. Therefore, policymakers and health planners should place strong emphasis on addressing the effectiveness, efficacy, and resistance management of LLINs as part of their current public health agenda to eliminate malaria.

## 1. Plain Language Summary

Malaria, a parasitic disease transmitted by mosquitoes, has been significantly reduced in recent decades through the use of LLINs. However, the rapid development of resistance to pyrethroids has halted this reduction. The World Health Organization recommends new LLINs with dual-active ingredients for areas with malaria vectors resistant to pyrethroids. This systematic review and meta-analysis aims to compare the effectiveness and efficacy of pyriproxyfen, chlorfenapyr, and piperonyl butoxide LLINs with pyrethroid-only LLINs and to evaluate the effectiveness and efficacy of chlorfenapyr and piperonyl butoxide LLINs compared to pyriproxyfen LLINs for malaria control in Africa. Critical appraisal of individual randomized controlled trials revealed that 100% of the studies scored as high quality, and Cochrane methodology was used to assess the risk of bias and evaluate evidence quality, which was graded as high. This research provides a very good indication of the likely effect. The likelihood that the effect will be substantially different is low.

A total of 11 cluster randomized controlled trials were conducted, involving 21,916 households, 1,145,035 individuals, and 34,327 children, as reported across all studies. This study found that the pooled prevalence of post-intervention malaria infection among children using chlorfenapyr, piperonyl butoxide, and pyriproxyfen LLINs was 25.58 per 100 children, 32.38 per 100 children, and 33.70 per 100 children, respectively, compared to the pyrethroid-only LLINs control group, which had a rate of 40.84% per 100 children in Africa.

This study found that the post-intervention pooled mean indoor vector density per household per night in the pyrethroid-only LLINs control group was higher than in the intervention groups, with pyrethroid-only nets having the highest density at 8.04 per household per night, compared to other insecticidal nets (7.74 per 100 households in pyriproxyfen, 5.53 per 100 households in chlorfenapyr, and the lowest, 1.9 per 100 households per night, in piperonyl butoxide) in Africa.

This study determined that the post-intervention pooled sporozoite rate per mosquito in the pyrethroid-only LLINs control group was almost two- to three-times higher than in the intervention groups, with pyrethroid-only nets having the highest sporozoite rate per mosquito at 227 per 100 anopheles, compared to other interventional LLINs (165 per 100 anopheles in pyriproxyfen, 172 per 100 anopheles in piperonyl butoxide, and the lowest, 79 per 100 anopheles, in chlorfenapyr) in Africa.

A meta-analysis found that pyriproxyfen (PPF) LLINs effectively reduce indoor vector density by 1%, the entomological inoculation rate by 7%, and the sporozoite rate of malaria parasites by 15% compared to pyrethroid-only LLINs in Africa.

This study found that piperonyl butoxide (PBO) LLINs are effective in reducing malaria infection by 1%, case incidence by 2%, and anemia by 3% among children, as well as reducing indoor vector density by 3%, the mean entomological inoculation rate by 12%, and the sporozoite rate by 10% in Africa, as compared to pyrethroid-only LLINs in Africa.

This study found that chlorfenapyr (CFP) LLINs are effective in reducing malaria infection by 1%, case incidence by 1%, and anemia by 4% among children, as well as reducing indoor vector density by 4%, the inoculation rate by 23%, and the sporozoite rate by 9% in Africa, as compared to pyrethroid-only LLINs.

This study compared the effectiveness and efficacy of chlorfenapyr (CFP) and pyriproxyfen LLINs in Africa. The results showed that CFP nets were highly effective and efficacious in reducing malaria infection, case incidence, indoor vector density, inoculation rate, and sporozoite rate by 1%, 15%, and 7%, respectively, compared to pyriproxyfen long-lasting insecticidal nets for malaria control in Africa.

The evidence evaluating the effectiveness and efficacy of piperonyl butoxide (PBO) compared with pyriproxyfen long-lasting insecticidal nets showed that piperonyl butoxide (PBO) LLINs reduce malaria infection by 0.0%, case incidence by 2% among children, indoor vector density by 4%, inoculation rate by 5%, and sporozoite rate by 1%, as compared to pyriproxyfen LLINs for malaria control in Africa.

We evaluated the effectiveness and efficacy of pyriproxyfen, chlorfenapyr, and piperonyl butoxide long-lasting insecticidal nets against the pyrethroid-only LLINs. This study found that PYR-only LLINs (control arm) had a higher pooled prevalence of malaria infection, case incidence, anemia, mean indoor vector density, inoculation rate, and sporozoite rate, as compared to the intervention group (PPF, CFP, and PBO LLINs).

This meta-analysis reveals that pyriproxyfen (PPF) long-lasting insecticidal nets (LLINs) have no significant difference in malaria infection, case incidence, or anemia reduction among children, as compared to pyrethroid-only LLINs. However, this study found that pyriproxyfen (PPF) LLINs effectively and efficaciously reduce indoor vector density, entomological inoculation rate, and sporozoite rate of malaria parasites compared to pyrethroid-only LLINs.

## 2. Introduction

Over the past 20 years, significant investment in long-lasting insecticidal nets, indoor residual spraying, and artemisinin-based therapies has led to the aversion of 1.5 billion malaria cases and 6.6 million deaths [1]. Long-lasting insecticidal nets (LLINs) are the foundation of malaria control, but the resistance of mosquito vectors to pyrethroids threatens their effectiveness [2]. The World Health Organization (WHO) reported that vector resistance to insecticides was observed in 88 malaria-endemic countries from 2010 to 2020 [3]. Globally, resistance to pyrethroids was reported in 64% of sites reporting resistance and, with high-intensity resistance, detected in 27 countries and at 293 sites. The monitoring of insecticide resistance has been adjusted to align with new procedures, including new discriminating concentrations and procedures for chlorfenapyr, clothianidin, transfluthrin, flupyradifurone, and pyriproxyfen [3]. Pyrethroid (PY) long-lasting insecticidal nets (LLINs) are the primary malaria control method in sub-Saharan Africa, contributing significantly to the decline in malaria morbidity and mortality. However, the rapid spread of PY resistance in vector populations has hindered this decline [4].

Long-lasting insecticidal nets’ (LLINs) effects may vary based on the type of vector of mosquito species [5]. Malaria prevalence reduction has been undertaken through the upgrading of pyrethroid long-lasting insecticidal nets (LLINs); however, the decrement in malaria burden stopped, which may related to pyrethroid fast resistance [6]. The World Health Organization described piperonyl butoxide (PBO) LLINs as more effective against malaria than non-PBO LLINs in the case of resistance to pyrethroids [2]. Evidence indicates that outdoors, no reduction in the density of malaria was observed in the pyrethroid pyriproxyfen LLIN arm [7].

Malaria control in sub-Saharan Africa faces threats from pyrethroid resistance, prompting the development of new long-lasting insecticidal nets (LLINs) with dual-active ingredients to interrupt transmission in pyrethroid-resistant areas [3]. A randomized controlled trial showed that chlorfenapyr LLINs are most effective against anopheles funestus over three years. PBO LLINs lasted two years. Anopheles arabiensis was unaffected, requiring additional interventions [8]. A study in Southern Mali found that deltamethrin + PBO LLINs reduced An. gambiae sporozoite rates during high malaria transmission seasons but no improvement in parity rates or indoor resting densities. Combination nets may be more effective in areas with mixed-function oxidases [9].

New LLINs combining insecticides with different modes of action may help restore malaria control in sub-Saharan Africa.

A cluster randomized trial in southern Benin showed chlorfenapyr–pyrethroid LLINs offer better malaria protection than pyrethroid-only nets. Pyriproxyfen–pyrethroid LLINs showed no added benefit. Further research is needed for improved vector control [10]. A third-year post-distribution secondary analysis of a cluster randomized controlled trial in southern Benin found that the pyriproxyfen–pyrethroid LLIN group did not provide superior protection against malaria cases compared to the standard LLIN group, and the chlorfenapyr–pyrethroid LLIN group also did not offer superior protection against malaria cases or infections [11]. A cluster randomized trial in Tanzania found that malaria incidence was consistently lower in too-torn PBO-PY LLIN and chlorfenapyr-PY LLIN compared to intact PY-only LLIN during the first year of follow-up. In year 2, the incidence was only significantly lower in intact chlorfenapyr-PY LLINs compared to intact PY LLINs [12].

A systematic review and meta-analysis found that chlorfenapyr–pyrethroid insecticide-treated nets are more effective in reducing malaria case incidence and parasite prevalence than pyrethroid-only ITNs. However, only chlorfenapyr-pyrethroid ITNs showed a reduction in these outcomes compared to pyrethroid-PBO ITNs [13]. These inconsistent approaches to measuring the effectiveness and efficacy of long-lasting insecticidal nets (LLINs) pose challenges to programmers and policymakers. Hence, it is important to examine the effectiveness and efficacy of long-lasting insecticidal nets (LLINs) by comparing different long-lasting insecticidal nets (LLINs) over a range of 6 months to 36 months post-intervention distribution of long-lasting insecticidal nets (LLINs). However, to the best of our knowledge, no systematic review or meta-analysis has been conducted to estimate the pooled effectiveness and efficacy of long-lasting insecticidal nets (LLINs). Therefore, considering the scarcity of evidence on the effectiveness and efficacy of pyriproxyfen, chlorfenapyr, and piperonyl butoxide long-lasting insecticidal nets (LLINs) with pyrethroid-only LLINs for malaria control, we aimed to fill this evidence gap. Hence, programmers and policymakers rely on the evidence from their businesses. Furthermore, researchers can gain insights into another research question to further study long-lasting insecticidal nets (LLINs). This systematic review and meta-analysis aimed to inform governments, policymakers, health professionals, and populations at risk of malaria infections of the effectiveness and efficacy of long-lasting insecticidal nets (LLINs) and to evaluate changes and trends in the effectiveness and efficacy of pyriproxyfen, chlorfenapyr, and piperonyl butoxide long-lasting insecticidal nets (LLINs) with pyrethroid-only LLINs, and this study also compares the effectiveness of chlorfenapyr and piperonyl butoxide LLINs compared to pyriproxyfen LLINs for malaria control over time.

### The Objectives of This Review

To conduct a systematic review of long-lasting insecticidal nets’ (LLINs) effectiveness and efficacy for malaria control.

To conduct a meta-analysis to estimate the effectiveness and efficacy of pyriproxyfen, chlorfenapyr, and piperonyl butoxide long-lasting insecticidal nets (LLINs) with pyrethroid-only LLINs for malaria control in Africa.

To compare the effectiveness and efficacy of pyriproxyfen, chlorfenapyr, and piperonyl butoxide LLINs with pyrethroid-only LLINs for malaria control in Africa.

To evaluate the effectiveness and efficacy of chlorfenapyr and piperonyl butoxide long-lasting insecticidal nets compared to pyriproxyfen LLINs for malaria control in Africa.

## 3. Methods

The protocol of this systematic review and meta-analysis was registered in PROSPERO (registration number: CRD42024499800, available from: https://www.crd.york.ac.uk/PROSPERO/view/CRD42024499800, accessed on 24 January 2024). 

This systematic review used Cochrane methodology for systematic reviews of interventional studies to assess risk of bias and evaluate evidence quality [14]. The analysis, interpretation, and reporting included a risk of bias assessment using the Cochrane Risk of Bias tool, which assigns studies as having a low, unclear, or high risk of bias. Quality of evidence was evaluated based on the Grades of Recommendation, Assessment, Development, and Evaluation (GRADE) approach, which involves consideration of methodological quality, directness of evidence, heterogeneity, precision of effect estimates, and publication bias [15].

### 3.1. Literature Search Strategy

A systematic literature search was conducted to identify relevant articles from online databases, like PubMed, MEDLINE, Embase, and the Cochrane Central Register of Controlled Trials’ database (CENTRAL), for retrieving randomized control trials comparing the effectiveness and efficacy of pyriproxyfen, chlorfenapyr, and piperonyl butoxide long-lasting insecticidal nets (LLINs) compared with pyrethroid-only LLINs for malaria control in Africa. The studies were searched from the following databases, without restriction on the date of publication among all age groups, and children aged 5 years or younger were eligible to receive interventions of long-lasting insecticidal nets (LLINs). MEDLINE, Hinari, Scopes, PubMed CINAHL, PopLine, MedNar, Embase, the Cochrane Library, JBI Library, Web of Science, and Google Scholar and reference lists of selected articles were also screened for identifying additional potentially eligible studies. The following is an example of a search string in PubMed: “” (Effectiveness and cost-effectiveness analysis [MeSH] OR cost effectiveness and against (malaria [MeSH] AND pyriproxyfen [MeSH] OR, chlorfenapyr [MeSH] OR, and piperonyl butoxide long-lasting insecticidal nets (LLINs) [MeSH] OR du-al-active-ingredient AND longlasting AND insecticidal AND nets AND LLINs AND (compared pyrethroid-only AND LLINs or (bednet OR bednets OR insecticide treated net OR insecticide treated nets OR insecticide treated bednet OR insecticide treated bednets OR insecticide treated bed net OR insecticide treated bed nets OR itn s OR itn OR itns OR llin* OR long-lasting insecticidal net OR long lasting insecticidal nets OR long lasting insecticide net OR long lasting insecticide nets OR mesh OR meshes OR net OR nets OR netting) Or Insecticides, Pyrethrins Piperonyl Butoxide) (Appendix A).

### 3.2. Study Eligibility

This review included randomized control trials or cluster randomized control trials or prospective clinical trials comparing long-lasting insecticidal nets (LLINs) of pyriproxyfen, chlorfenapyr, and/or piperonyl butoxide for malaria control (test arm) and pyrethroid-only standard LLINs (control arm) for malaria control. This study involved all studies including children aged 6 months to 14 years, permanently residing in a selected household, and an adult caregiver using long-lasting insecticidal nets (LLINs) for malaria control.

### 3.3. Participants

Studies conducted on adults and children who are residents of a region with ongoing malaria transmission and were provided with long-lasting insecticidal nets (LLINs) in Africa were eligible for this review.

### 3.4. Study Design

Only cluster randomized and non-randomized cluster-controlled studies that included more than one cluster per arm were considered for this review. Non-randomized controlled study designs were only considered for inclusion when there was a comparison/control group present. There were no exclusion rules based on any buffer period (i.e., when participants act as their own controls) or length of intervention or timing of measurement of outcomes. All observational studies and modelling studies were excluded.

### 3.5. Intervention(s)

The interventions of interest are all studies assessing the effectiveness or efficacy of pyriproxyfen, chlorfenapyr, and piperonyl butoxide long-lasting insecticidal nets (LLINs) compared to the standard care (pyrethroid-only long-lasting insecticidal nets (LLINs) for malaria control in context of Africa).

### 3.6. Comparator(s)/Control

The standard care of pyrethroid-only long-lasting insecticidal nets (LLINs) treated for malaria control in context of Africa was applied.

### 3.7. Main Outcome(s)

The outcome measures for this systematic review were based on the effectiveness and efficacy of long-lasting insecticidal nets (LLINs) on malaria control. Key outcomes included malaria infection prevalence in children, anemia prevalence in children aged 6 months to 4 years, entomological indicators (mean indoor vector density and entomological inoculation rate per household per night), malaria case incidence in children aged 6 months to 10 years, overall vector density, and sporozoite rate. Efficacy and effectiveness were estimated as the proportion of malaria prevention outcomes among users of pyrethroid-only LLINs, calculated as the number of positive outcomes divided by the total number of participants, multiplied by 100. Additionally, factors associated with efficacy, effectiveness, and malaria prevalence will be reported as risk differences, based on binary outcomes from the original studies.

### 3.8. Operational Definition of Outcomes

#### 3.8.1. Primary Outcome

The primary outcome was the prevalence of malaria infection, defined as a positive rapid diagnostic test (RDT) in children aged 6 months to 14 years, measured 24 months after the distribution of long-lasting insecticidal nets (LLINs). Secondary assessments of this outcome were conducted at 12 and 18 months.

#### 3.8.2. Malaria Infection Incidence

Malaria infection incidence was defined as parasitemia with or without symptoms, measured over a population at risk or person-time and detected through passive or active surveillance.

#### 3.8.3. Secondary Outcomes

Malaria Case Incidence: Defined as fever (≥37.5 °C or within the past 48 h) plus a positive RDT in children aged 6 months to 10 years, measured over 24 months.

Severe Malaria Incidence: Defined as hospitalization with parasitemia, measured over a population at risk or person-time.

Malaria Case Incidence Rate: Defined as the presence of malaria symptoms and parasitemia, measured over a population at risk or person-time.

Anemia Prevalence: Based on study-defined thresholds, including moderate to severe anemia (hemoglobin <8 g/dL) in children aged 6 months to 4 years, measured at 12, 18, and 24 months.

Malaria Transmission: Measured using the entomological inoculation rate (EIR), defined as the number of mosquito vectors testing positive for malaria over 24 months.

#### 3.8.4. Entomological Outcomes

Entomological outcomes were included only when epidemiological data were also reported. They were extracted but not used as primary outcomes:

EIR: Number of infective bites per person per unit time.

Sporozoite Rate: Percentage of female Anopheles mosquitoes with sporozoites in their salivary glands.

Anopheline Density: Number of female Anopheles mosquitoes per shelter, host, or sampling period, with collection methods specified.

Vector Density: Mean number of malaria vectors collected per house per night.

##### Additional Timepoints

Secondary outcomes, including malaria prevalence, anemia, and EIR, were also assessed at 30 and 36 months post-intervention [13,16].

### 3.9. Setting

Studies conducted in countries with ongoing malaria transmission were considered for this review. The presence of other background interventions did not impact study eligibility if they were present in both arms equally. Studies where additional malaria interventions are considered standard of care were included if interventions (both malaria and non-malaria) were balanced between intervention and control arms. All identified and retrieved studies were from Africa only.

### 3.10. Data Extraction

Data were extracted independently by two reviewers using a standardized form with pre-defined criteria, as set out by the protocol, and checked and verified by a third reviewer. It included cluster randomized or prospective clinical trials comparing long-lasting insecticidal nets (LLINs) of pyriproxyfen, chlorfenapyr, and/or piperonyl butoxide for malaria control (test arm) and pyrethroid-only standard LLINs (control arm) for malaria control. The criteria according to which data were extracted comprised the following: participant demographics (mean age, household, total number of clusters, total population, number of selected children, malaria infection and anemia prevalence in selected children); intervention (long-lasting insecticidal nets (LLINs) of pyriproxyfen, chlorfenapyr, and/or piperonyl butoxide) and comparator details (pyrethroid-only standard LLINs for malaria control intervention); design features (cluster randomized or prospective clinical trials [cRCT] design comparing long-lasting insecticidal nets, study setting, malaria infection and case incidence, entomological and vector density, and sporozoite rate outcome measures); and outcomes (post-treatment risk ratio (RR) with 95% confidence interval (CI) for malaria controls and other interventions (long-lasting insecticidal nets (LLINs) treatments at the end of treatment and follow-up)).

### 3.11. Data Synthesis and Statistical Analysis

The data were reported in accordance with Preferred Reporting of Systematic Reviews and Meta-Analyses (PRISMA), with the updated guidance of the RISMA 2020 statement [17]. We verified the appropriateness of each datum prior to analysis. Pooled estimates were calculated using the STATA Version 17 software (STATA Corporation, College Station, TX, USA). The extracted data from eligible studies were pooled using the random effects model and expressed as a risk ratio (RR) with a 95% confidence interval (CI). Both random- and fixed-impact methods were used to measure the pooled estimates. The pooled estimates were computed using “metaprop” using a sample size as a weight (wgt) variable with 95% CIs. Pooled estimates were computed using random-effects models and weighted using the inverse variance method in the presence of high heterogeneity among studies.

Heterogeneity was assessed using the *Q*, *I*^2^ and Tau (τ) statistics. The *I*^2^ statistic was used to indicate the percentage of overall variability attributable to between-study heterogeneity, categorized into low (25%), moderate (50%), and substantial (75%) groupings according to guidelines reported by Higgins [18]. Tau was reported to provide a robust estimate of the variance in true effect sizes (*SD*), which is not susceptible to influences from number and precision of included studies (as can be the case for *Q* and *I*^2^). Any *p*-value < 0.05 was considered as statistically significant. Sensitivity analysis, subgroup analysis, and publication bias were also assessed as appropriate. Subgroup analyses were performed using different parameters (country, AOR, intervention types, duration of follow-up). Forest plots, summary tables, and text are used to present the findings of this study.

### 3.12. Quality Assessment

The quality of the study was assessed using the revised Joanna Briggs Institute (JBI) critical appraisal tool for the assessment of risk of bias for randomized controlled trials [19], and the results were graded as low, medium, or high if the quality score was <60%, 60–80%, or >80%, respectively. We inspected the funnel plot and conducted Egger’s regression tests to assess publication bias [20] (Appendix A).

### 3.13. Publication Bias

The likelihood of publication bias was evaluated using four methods to prevent overreliance on one approach. These comprised the following: (1) funnel plots of standardized mean differences plotted against standard errors, which were visually observed to detect possible asymmetry; (2) [21] Trim and Fill imputation was used to predict the adjusted combined effect size (ES) taking account of publication bias; (3) Egger’s regression was utilized as a formal statistical assessment of potential publication bias by regressing standardized effect estimates onto a measure of precision [22,23]; and (4) Rosenthal fail-safe *N* calculation, which estimates the number of additional studies with an ES of zero required to turn the overall effect insignificant [24]. In addition, Egger’s weighted regression and Begg’s tests will be used to check publication bias. The statistical significance of publication bias was declared at a *p*-value of less than 0.05.

### 3.14. GRADE Analysis

In the meta-analysis, the quality of the evidence was assessed for each comparison using the Grading of Recommendations Assessment, Development, and Evaluation (GRADE) tool [25]. Meta-analytic comparisons were graded by non-reviewed authors to reach a consensus quality rating (high, moderate, low, or very low quality) based on five domains: (1) risk of bias in the individual included studies, (2) inconsistency, (3) indirectness of treatment estimate effects, (4) imprecision, and (5) publication bias (Evidence Grading Appendix A).

## 4. Results

### 4.1. Selection of Studies

The initial search yielded 952 studies from databases and grey literature sources. After removing 409 duplicates, 543 studies remained. These were screened by title and abstract, resulting in the exclusion of 487 studies. The full texts of the remaining 56 studies were then assessed for eligibility [2,6,8,9,10,11,12,16,26,27,28,29,30,31,32,33,34,35,36,37,38,39,40,41,42,43,44,45,46,47,48,49,50,51,52,53,54,55,56,57,58,59,60,61,62,63,64,65,66,67,68,69,70,71,72,73]. Of the 56 full-text studies assessed, 39 were excluded due to inconsistent results and study designs, and 6 were study protocol-only. Ultimately, 11 studies met the inclusion criteria and were included in the final systematic review and meta-analysis [2,6,8,10,11,12,16,27,28,29,30] (see details in Figure 1).

### 4.2. Study Characteristics

This review included 11 studies comprising data from over 21,916 households, 1,145,035 individuals in core and buffer areas, and 34,327 children, based on reported sample sizes. All studies were conducted in Africa. Of these, six were from Tanzania [6,8,12,16,28,29], two each from Benin [10,11] and Uganda [2,30], and one from Kenya [27] (Appendix A).

In summary, these studies were conducted in four African countries, and all studies were cluster randomized control trials or randomized control trials. Critical appraisal of randomized control trials revealed that 100% of the studies scored high quality, and Cochrane methodology was used to assess the risk of bias and evaluate evidence quality, which was graded as high. This research provides a very good indication of the likely effect. The likelihood that the effect will be substantially different is low (Evidence Grading Appendix A).

#### 4.2.1. Pooled Baseline Malaria Infection Prevalence in Selected Children Using Different Long-Lasting Insecticidal Nets (LLINs) as Malaria Control in Africa

##### Pooled Effectiveness and Efficacy of Pyriproxyfen Long-Lasting Insecticidal Nets (LLINs) Versus Pyrethroid-Only LLINs Malaria Infection Reduction

Forest plots showed that malaria infection risk reduction among children using the pyriproxyfen intervention group versus pyrethroid-only long-lasting insecticidal nets (LLINs) standard treatment revealed no significant difference (RR = −0.00 with 95% CI −0.04, 0.03). Figure 1 reveals that there is no significant difference in malaria prevalence reduction between the treatment and control groups. Furthermore, the included studies were homogeneous, with an *I*^2^ of 0.00. See details in Figure 2.

##### Pooled Prevalence of Malaria Infection Among Children Using Pyriproxyfen Long-Lasting Insecticidal Nets (LLINs) Versus Pyrethroid-Only LLINs

This randomized control trial meta-analysis found the estimated pooled prevalence of malaria infection among children using pyriproxyfen long-lasting insecticidal nets (LLINs) was 42.8 per 100 children (95% CI: 39.47–46.12%) and 47.80% (37.77%, 57.84%) in the pyrethroid-only LLINs control group (see details in Appendix A).

##### Pooled Effectiveness and Efficacy of Chlorfenapyr Long-Lasting Insecticidal Nets (LLINs) Versus Pyrethroid-Only LLINs Malaria Infection Reduction

Forest plots showed that malaria infection risk reduction among children using chlorfenapyr long-lasting insecticidal nets (LLINs) reduced the risk of malaria infection by 1% compared to standard or pyrethroid-only LLINs (RR = −0.01 with a 95% CI of −0.04–0.03) (see details in Appendix A).

##### Pooled Prevalence of Malaria Infection Among Children Using Chlorfenapyr Long-Lasting Insecticidal Nets (LLINs) Versus Pyrethroid-Only LLINs

This meta-analysis of randomized control trials found that the pooled prevalence of malaria infection among children using chlorfenapyr long-lasting insecticidal nets was 38.09 per 100 (95% CI: 39.47–46.12%), while pyrethroid-only LLINs had a nearly 50% (47.80 with 95% CI 37.77%, 57.84%) pooled prevalence of malaria infection among children using pyrethroid-only long-lasting insecticidal nets (LLINs) (see details in Appendix A).

##### Pooled Effectiveness and Efficacy of Piperonyl Butoxide Long-Lasting Insecticidal Nets (LLINs) Versus Pyrethroid-Only LLINs Malaria Infection Reduction

Forest plots showed that malaria infection risk reduction among children using the piperonyl butoxide versus pyrethroid-only long-lasting insecticidal nets (LLINs) standard treatment revealed no significant difference (RR = −0.00 with 95% CI −0.03, 0.02).

This study shows no significant difference in malaria prevalence reduction between the treatment and control groups, and the included studies were homogeneous, with an *I*^2^ of 0.00. See details in Appendix A.

##### Pooled Prevalence of Malaria Infection Among Children Using Piperonyl Butoxide Long-Lasting Insecticidal Nets (LLINs) Versus Pyrethroid-Only LLINs

This randomized control trial meta-analysis determined the estimated pooled prevalence of malaria infection among children using Piperonyl butoxide long-lasting insecticidal nets (LLINs) were 43.91 per 100 children with (95% CI: 33.43%, 54.39%), while pyrethroid-only LLINs had a nearly 50% (47.80 with 95% CI 37.77%, 57.84%) pooled prevalence malaria infection among children using pyrethroid-only long-lasting insecticidal nets (LLINs) (see details in Appendix A).

##### Pooled Prevalence of Malaria Infection Among Children Using Pyrethroid-Only Long-Lasting Insecticidal Nets (LLINs)

This randomized control trial meta-analysis determined the estimated pooled prevalence of malaria infection among children using pyrethroid-only long-lasting insecticidal nets (LLINs) was 47.80 per 100 children (95% CI: 37.77%, 57.84%) in Africa (see details Appendix A).

This funnel plot indicates that there is no publication bias among the included studies as it is symmetrically distributed. Hence, the funnel plot seems symmetric, and Egger’s regression test (PV= 0.1460) confirmed no publication bias among the included studies (Appendix A).

#### 4.2.2. Pooled Baseline Anemia Prevalence in Children Aged 6 Months to 4 Years Using Different Long-Lasting Insecticidal Nets (LLINs) as Malaria Control in Africa

##### Pooled Effectiveness and Efficacy of Pyriproxyfen Long-Lasting Insecticidal Nets (LLINs) Versus Pyrethroid-Only LLINs for Anemia Reduction in Children 

Forest plots showed no difference in anemia risk reduction among children aged 6 months to 4 years using pyriproxyfen long-lasting insecticidal nets versus the pyrethroid-only LLINs standard treatment, and the included studies were homogeneous (*I*^2^ = 0.01%). See details in Appendix A.

##### Pooled Baseline Anemia Prevalence in Children Aged 6 Months to 4 Years Using Pyriproxyfen Long-Lasting Insecticidal Nets (LLINs) Versus Pyrethroid-Only LLINs

This randomized control trial meta-analysis found the estimated pooled prevalence of anemia among children aged 6 months to 4 years using pyriproxyfen long-lasting insecticidal nets (LLINs) was 29.28 per 100 children (95% CI: 5.81–52.75%) and 25.18% (12.78%, 37.58%) in the pyrethroid-only LLINs control group (see details in Appendix A).

##### Pooled Effectiveness and Efficacy of Chlorfenapyr Long-Lasting Insecticidal Nets (LLINs) Versus Pyrethroid-Only LLINs for Anemia Reduction in Children 

Forest plots showed that anemia risk among children aged 6 months to 4 years using chlorfenapyr long-lasting insecticidal nets (LLINs) was reduced by 1% compared to standard or pyrethroid-only LLINs (RR = −0.01 with a 95% CI of −0.05–0.03). See details in Appendix A.

##### Pooled Prevalence of Anemia Among Children Using Chlorfenapyr Long-Lasting Insecticidal Nets (LLINs) Versus Pyrethroid-Only LLINs

This study found that the pooled prevalence of anemia among children aged 6 months to 4 years using chlorfenapyr long-lasting insecticidal nets was 29.36 per 100 children, compared to 25.18% in the control/pyrethroid-only group (see details in Appendix A).

##### Pooled Effectiveness and Efficacy of Piperonyl Butoxide Long-Lasting Insecticidal Nets (LLINs) Versus Pyrethroid-Only LLINs for Anemia Reduction in Children 

Forest plots showed that anemia risk among children aged 6 months to 4 years using piperonyl butoxide long-lasting insecticidal nets (LLINs) was reduced by 2% compared to standard or pyrethroid-only LLINs (RR = −0.02 with a 95% CI of −0.07, 0.04). See details in Appendix A.

##### Pooled Prevalence of Anemia Among Children Using Piperonyl Butoxide Long-Lasting Insecticidal Nets (LLINs) Versus Pyrethroid-Only LLINs

This study found that the pooled prevalence of anemia among children aged 6 months to 4 years using piperonyl butoxide long-lasting insecticidal nets was 14.31 per 100 children, compared to 25.18% in the control/pyrethroid-only group (see details in Appendix A).

##### Pooled Prevalence of Anemia Among Children Using Pyrethroid-Only Long-Lasting Insecticidal Nets (LLINs)

This study found that the pooled prevalence of anemia among children aged 6 months to 4 years using pyrethroid-only long-lasting insecticidal nets (LLINs) was 25.18 per 100 children (95% CI: 12.78%, 37.58%) in Africa 2024 (see details in Appendix A).

##### Baseline Pooled Mean Indoor Vector Density per Household per Night

This meta-analysis determined entomological outcomes including effectiveness and efficacy in terms of mean indoor vector density per household per night reduction by using different long-lasting insecticidal nets (LLINs) as malaria control in Africa.

Forest plots showed no difference in mean indoor vector density per household per night reduction using pyriproxyfen long-lasting insecticidal nets versus pyrethroid-only LLINs standard treatment, and the included studies were homogeneous (*I*^2^ = 0.00%). This indicates no additional effectiveness or efficacy in the mean indoor vector density in the study area. See details in Appendix A.

##### Pooled Prevalence of Mean Indoor Vectors Density per Household per Night Using Pyriproxyfen Long-Lasting Insecticidal Nets (LLINs)

This meta-analysis found that pyriproxyfen long-lasting insecticidal nets (LLINs) did not significantly reduce indoor vector density per household per night compared to the standard/control group. The pooled prevalence of mean indoor vector density was 11.28 per household per night in pyriproxyfen LLINs, compared to 12.78 per household per night in pyrethroid-only LLINs (see details in Appendix A).

##### Pooled Prevalence of Mean Indoor Vectors Density per Household per Night Using Chlorfenapyr Long-Lasting Insecticidal Nets (LLINs)

A forest plot shows the effectiveness and efficacy of the chlorfenapyr long-lasting insecticidal nets (LLINs) intervention, which decreases the risk of mean indoor vector density per household per night by 2% when controlled with the standard care/pyrethroid-only LLINs in Africa in 2024. See details Appendix A.

##### Pooled Prevalence of Mean Indoor Vectors Density per Household per Night Using Chlorfenapyr Long-Lasting Insecticidal Nets (LLINs)

The pooled prevalence of mean indoor vector density was 6.51 per household per night in chlorfenapyr LLINs, compared to 12.78 per household per night in pyrethroid-only LLINs (see details in Appendix A).

The effectiveness and efficacy of piperonyl butoxide long-lasting insecticidal nets (LLINs) showed no difference in reducing mean indoor vector density per household per night compared to pyrethroid-only LLINs

The forest plot demonstrates that piperonyl butoxide LLINs significantly reduce mean indoor vector density in Africa by 2% compared to standard care/pyrethroid-only LLINs in 2024. See details in Appendix A.

##### Pooled Prevalence of Mean Indoor Vectors Density per Household per Night Using Piperonyl Butoxide LLINs

The pooled prevalence of mean indoor vector density was 13.49 per household per night in piperonyl butoxide LLINs compared to 12.78 per household per night in pyrethroid-only LLINs (see details in Appendix A).

##### Pooled Prevalence of Mean Indoor Vectors Density per Household per Night Using Pyrethroid-Only LLINs

This study found that the pooled prevalence of mean indoor vector density using pyrethroid-only long-lasting insecticidal nets (LLINs) was 12.78 per household per night (95% CI: 5.69, 19.87%) in Africa in 2024 (see details in Appendix A).

#### 4.2.3. Baseline Sporozoite Rate Is the Proportion of Vectors Infected with Malaria Parasite

##### Pooled Effectiveness and Efficacy of Pyriproxyfen Long-Lasting Insecticidal Nets (LLINs) Versus Pyrethroid-Only LLINs for Sporozoite Rate Reduction

Forest plots showed that pyriproxyfen intervention effectively reduces the sporozoite rate by 2% compared to standard or pyrethroid-only LLINs (RR = −0.02 with a 95% CI of −0.37, 0.33). See details in Appendix A.

##### Pooled Prevalence of Sporozoite Rate Among Pyriproxyfen LLINs Intervention

The pooled prevalence of the sporozoite rate was 3.30 in the pyriproxyfen LLIN intervention compared to 4.57 in the pyrethroid-only LLINs standard (see details in Appendix A).

##### Pooled Effectiveness and Efficacy of Chlorfenapyr Long-Lasting Insecticidal Nets (LLINs) Versus Pyrethroid-Only LLINs for Sporozoite Rate Reduction

This study found that chlorfenapyr intervention effectively reduces the sporozoite rate by 5%, as compared to standard or pyrethroid-only LLINs, as shown in Appendix A.

##### Pooled Prevalence of Sporozoite Rate Among Chlorfenapyr LLINs Intervention

The pooled prevalence of the sporozoite rate was 2.20 in the chlorfenapyr LLIN intervention compared to 4.57 in the pyrethroid-only LLINs standard (see details in Appendix A).

##### Pooled Prevalence of Sporozoite Rate Among Piperonyl Butoxide LLINs Intervention

This study found that piperonyl butoxide intervention effectively reduces the sporozoite rate by 2%, as compared to standard or pyrethroid-only LLINs, as shown in Appendix A.

##### Pooled Prevalence of Sporozoite Rate Among Piperonyl Butoxide LLINs Intervention

The pooled prevalence of the sporozoite rate was 3.50 in piperonyl butoxide LLIN intervention compared to 4.57 in the pyrethroid-only LLINs standard (see details in Appendix A).

##### Pooled Prevalence of the Sporozoite Rate Among Pyrethroid-Only LLINs Intervention

The pooled prevalence of the sporozoite rate was 4.57 in the pyrethroid-only LLINs standard (see details in Appendix A).

##### Mean Entomological Inoculation Rate per Household per Night (MEIR)

This meta-analysis determined entomological outcomes including effectiveness and efficacy in terms of the mean entomological inoculation rate per household per night (MEIR) reduction and reported that the pyriproxyfen LLINs intervention was reduced by 8%, as compared to the standard or control of the pyrethroid-only group (AOR = −0.08, 95% CI = −1.03, 0.87). See details in Appendix A.

This study determined the pooled mean entomological inoculation rate per night (23 per 100 households per night with a 95% CI of 0.01, 0.48) in the pyriproxyfen LLINs intervention (versus 56 per 100 households per night in the pyrethroid-only group, with a 95% CI of 0.23, 0.88) (see details in Appendix A).

This meta-analysis determined entomological outcomes including effectiveness and efficacy in terms of the mean per household per night inoculation rate and reported that the chlorfenapyr group was reduced by 12%, as compared to the standard or control of the pyrethroid-only group (AOR = −0.12, 95% CI = −1.21, 0.97). See details in Appendix A.

This study determined the pooled mean entomological inoculation rate per night (8 per 100 households per night with a 95% CI of (−0.03, 0.20) in the chlorfenapyr LLINs intervention (versus 56 per 100 households per night in the pyrethroid-only group, with a 95% CI of 0.23, 0.88) (see details in Appendix A).

This meta-analysis determined entomological outcomes including effectiveness and efficacy in terms of mean per household per night inoculation rate and reported that the piperonyl butoxide group was reduced by 21%, as compared to the standard or control of the pyrethroid-only group (AOR = −0.21, 95% CI = −1.95, 1.53). See details in Appendix A.

This study determined the pooled mean entomological inoculation rate per night (7 per 100 households per night with a 95% CI of (−0.10, 0.24) in the piperonyl butoxide LLINs intervention (versus 56 per 100 households per night in the pyrethroid-only group with a 95% CI of 0.23, 0.88) (see details in Appendix A).

This study determined the pooled mean entomological inoculation rate per night (56 per 100 households per night in the standard or control of the pyrethroid-only group with a 95% CI of 0.23, 0.88) (see details in Appendix A).

##### Pooled Post-Intervention Effectiveness and Efficacy of Pyriproxyfen Long-Lasting Insecticidal Nets (LLINs) Versus Pyrethroid-Only LLINs for Malaria Infection Reduction in Africa

This study evaluated the post-intervention effectiveness and efficacy of pyriproxyfen long-lasting insecticidal nets (LLINs) in malaria infection risk reduction in children over different durations (6, 12, 18, 24, and 36 months), as compared to placebo/standard pyrethroid-only LLINs, which revealed no difference in reducing the risk of infection with (RR = −0.00 with 95% CI −0.03, 0.02). See details in Appendix A. 

The subgroup analysis of post-intervention follow-up effectiveness and efficacy of pyriproxyfen long-lasting insecticidal nets (LLINs) showed a significant reduction in malaria infection among children over different durations, with a 1% reduction at 12 months and 1% at 36 months post-distribution of LLINs, compared to pyrethroid-only LLINs (ARR = −0.01, 95% CI = −0.08, 0.08) but with no difference in overall effectives and efficacy.

##### Pooled Post-Intervention Malaria Infection Prevalence in Selected Children Using Different Long-Lasting Insecticidal Nets (LLINs) as Malaria Control in Africa

This randomized control trial meta-analysis found that the pooled prevalence of post-intervention malaria infection among children using pyriproxyfen long-lasting insecticidal nets (LLINs) over different durations (6, 12, 18, 24, and 36 months) was 33.70 per 100 children (95% CI: 28.03–39.37%), while it was higher in the control group/pyrethroid-only LLINs (40.84% per 100 children) (95% CI: 32.45%, 49.22%) in Africa (see details in Appendix A).

Publication bias was checked using funnel plots looking at symmetrical distribution, and it was objectively verified using Egger’s regression test, which revealed that there was no publication bias (*p* < 0.174) (Appendix A).

##### Pooled Post-Intervention Effectiveness and Efficacy of Chlorfenapyr Long-Lasting Insecticidal Nets (LLINs) Versus Pyrethroid-Only LLINs for Malaria Infection Reduction in Africa

This study determined that the post-intervention effectiveness and efficacy of chlorfenapyr long-lasting insecticidal nets (LLINs) in malaria infection risk reduction in children over different durations (6, 12, 18, 24, and 36 months) were reduced by 1%, as compared to the standard or control of the pyrethroid-only group (ARR = −0.01, 95% CI = −0.04, 0.02). See details in Appendix A.

The subgroup analysis of the post-intervention follow-up effectiveness and efficacy of chlorfenapyr long-lasting insecticidal nets (LLINs) showed a significant reduction in malaria infection among children over different durations, with a 2% reduction at 12 months and 1% at 36 months post-distribution of LLINs, compared to pyrethroid-only LLINs (ARR = −0.01, 95% CI = −0.04, 0.02). See details in Appendix A.

##### Pooled Post-Intervention Malaria Infection Prevalence in Selected Children Using Chlorfenapyr Long-Lasting Insecticidal Nets (LLINs) as Malaria Control in Africa

This randomized control trial meta-analysis found that the pooled prevalence of post-intervention malaria infection among children using chlorfenapyr long-lasting insecticidal nets (LLINs) over different durations (6, 12, 18, 24, and 36 months) was 25.58 per 100 children (95% CI: 19.52–31.64%), while it was doubled in the control group/pyrethroid-only LLINs (40.84% per 100 children) (95% CI: 32.45%, 49.22%) in Africa (see details in Appendix A).

Publication bias was checked using funnel plots looking at symmetrical distribution, and it was objectively verified using Egger’s regression test, which revealed that there was no publication bias (*p* < 0.9125) (Appendix A).

##### Pooled Post-Intervention Effectiveness and Efficacy of Piperonyl Butoxide Long-Lasting Insecticidal Nets (LLINs) Versus Pyrethroid-Only LLINs for Malaria Infection Reduction in Africa

This study determined that the post-intervention effectiveness and efficacy of piperonyl butoxide long-lasting insecticidal nets (LLINs) in malaria infection risk reduction in children over different durations (6, 12, 18, 24, and 36 months) were reduced by 1%, as compared to the standard or control of the pyrethroid-only group (ARR = −0.01, 95% CI = −0.02, 0.01). See details in Appendix A.

The subgroup analysis of the post-intervention follow-up effectiveness and efficacy of piperonyl butoxide long-lasting insecticidal nets (LLINs) showed a significant reduction in malaria infection among children over different durations, with a 2% reduction at 9 months and 1% at 36 months post-distribution of LLINs, compared to pyrethroid-only LLINs (ARR = −0.01, 95% CI = −0.02, 0.01). See details in Appendix A.

##### Pooled Post-Intervention Malaria Infection Prevalence in Selected Children Using Piperonyl Butoxide Long-Lasting Insecticidal Nets (LLINs) in Africa

This randomized control trial meta-analysis found that the pooled prevalence of post-intervention malaria infection among children using piperonyl butoxide long-lasting insecticidal nets (LLINs) over different durations (6, 12, 18, 24, and 36 months) was 32.38 per 100 children (95% CI: 25.27–39.50%), while it was slightly higher in the control group/pyrethroid-only LLINs (40.84% per 100 children) (95% CI: 32.45%, 49.22%) in Africa (see details in Appendix A).

Publication bias was checked using funnel plots looking at symmetrical distribution, and it was objectively verified using Egger’s regression test, which revealed that there was no publication bias (*p* < 0.847) (Appendix A).

##### Pooled Post-Intervention Malaria Infection Prevalence in Selected Children Using Pyrethroid-Only Long-Lasting Insecticidal Nets (LLINs) or Control Group in Africa

This randomized control trial meta-analysis found that the pooled prevalence of post-intervention malaria infection among children using pyrethroid-only LLINs or control group was 40.84% per 100 children (95% CI: 32.45%, 49.22%) in Africa (see details in Appendix A).

Publication bias was checked using funnel plots looking at symmetrical distribution, and it was objectively verified using Egger’s regression test, which revealed that there was no publication bias (*p* < 0.066) (Appendix A).

##### Pooled Post-Intervention Effectiveness and Efficacy of Pyriproxyfen Long-Lasting Insecticidal Nets (LLINs) Versus Pyrethroid-Only LLINs Malaria Case Incidence Reduction in Children (Aged 6 Months to 10 Years in Africa

This study assessed the post-intervention effectiveness and efficacy of pyriproxyfen long-lasting insecticidal nets (LLINs) in reducing malaria case incidence in children aged 6 months to 10 years over various durations compared to placebo/standard pyrethroid-only LLINs. The results showed no significant difference in reducing malaria case incidence (RR = −0.00). See details in Appendix A.

##### Pooled Post-Intervention Malaria Case Incidence Reduction in Children (Aged 6 Months to 10 Years Using Pyriproxyfen Long-Lasting Insecticidal Nets (LLINs) Versus Pyrethroid-Only LLINs in Africa

This randomized control trial meta-analysis found that the pooled post-intervention malaria case incidence among children (aged 6 months to 10 years using pyriproxyfen long-lasting insecticidal nets (LLINs) over different durations (12, 24, and 36 months)) was 69 per 100 children years (95% CI: 0.46, 0.89), and the control group/pyrethroid-only LLINs reached 46 per 100 children years (95% CI: 0.28, 0.63) in Africa (see details in Appendix A).

##### Pooled Post-Intervention Effectiveness and Efficacy of Chlorfenapyr Long-Lasting Insecticidal Nets (LLINs) Versus Pyrethroid-Only LLINs Malaria Case Incidence Reduction in Children Aged 6 Months to 10 Years in Africa

This study evaluated the post-intervention effectiveness and efficacy of chlorfenapyr long-lasting insecticidal nets (LLINs) in malaria case incidence reduction in children aged 6 months to 10 years over different durations, 1 year to 2 years, and overall reduction by 4% as compared to the standard or control of the pyrethroid-only group (ARR = −0.04, 95% CI = −0.33, 0.26). See details in Appendix A.

This randomized control trial meta-analysis found that the pooled post-intervention malaria case incidence among children (aged 6 months to 10 years using chlorfenapyr long-lasting insecticidal nets (LLINs) and control group/pyrethroid-only LLINs over different durations (12, 24, and 36 months)) was the same at 46 per 100 children years (95% CI: 0.28, 0.63) in Africa (see details in Appendix A).

##### Pooled Post-Intervention Effectiveness and Efficacy of Piperonyl Butoxide Long-Lasting Insecticidal Nets (LLINs) Versus Pyrethroid-Only LLINs Malaria Case Incidence Reduction in Children Aged 6 Months to 10 Years in Africa

This study determined that the post-intervention effectiveness and efficacy of piperonyl butoxide long-lasting insecticidal nets (LLINs) in malaria case incidence reduction in children aged 6 months to 10 years over different durations, 1 year to 2 years, was reduced by 3% as compared to the standard or control of the pyrethroid-only group (ARR = −0.03, 95% CI = −0.57, 0.5). See details in Appendix A.

This randomized control trial meta-analysis found that the pooled post-intervention malaria case incidence among children (aged 6 months to 10 years using piperonyl butoxide long-lasting insecticidal nets (LLINs) over different durations (12, 24, and 36 months)) was 31 per 100 children years (95% CI: 0.19, 0.43), and the control group/pyrethroid-only LLINs reached 46 per 100 children years) (95% CI: 0.28, 0.63) in Africa (see details in Appendix A).

This randomized control trial meta-analysis found that the pooled prevalence of post-intervention malaria case incidence among children (aged 6 months to 10 years using pyrethroid-only LLINs or control standard care over different durations (12, 24, and 36 months)) was 46 per 100 children years (95% CI: 0.28, 0.63) in Africa (see details in Appendix A).

##### Post-Intervention: Pooled Mean Indoor Vector Density per Household per Night

This meta-analysis determined entomological outcomes including effectiveness and efficacy in terms of mean indoor vector density per household per night reduction by using different long-lasting insecticidal nets (LLINs) in Africa.

The forest plot demonstrates that pyriproxyfen long-lasting insecticidal nets significantly reduce pooled mean indoor vector density in Africa by 1% as compared to standard care/pyrethroid-only LLINs treatment (ARR = 0.01, 95% CI = −0.05, 0.08), and the included studies were homogeneous (*I*^2^ = 0.00%). See details in Appendix A.

##### Post-Intervention Pooled Mean Indoor Vector Density per Household per Night Among Pyriproxyfen Long-Lasting Insecticidal Nets Intervention

This study revealed that the pooled mean indoor vector density per household per night in the pyriproxyfen LLIN intervention was 7.74, which was slightly lower than the pyrethroid-only LLINs or control groups, with a mean of 8.04 per household per night, as shown in Appendix A.

Publication bias was checked using funnel plots looking at symmetrical distribution, and it was objectively verified using Egger’s regression test, which revealed that there was no publication bias (*p* < 0.847) (Appendix A).

This meta-analysis found that chlorfenapyr long-lasting insecticidal nets effectively and efficiently reduced mean vector density per household per night by 4% compared to pyrethroid-only LLNs, demonstrating their effectiveness and efficacy over varying durations from one to three years post-distribution in Africa (Appendix A).

##### Post-Intervention Pooled Mean Indoor Vector Density per Household per Night Among Chlorfenapyr Long-Lasting Insecticidal Nets

This study revealed that the pooled mean indoor vector density per household per night in the chlorfenapyr long-lasting insecticidal nets intervention was 5.53, which was significantly lower than the pyrethroid-only LLINs or control groups, with a mean of 8.04 per household per night, as shown in Appendix A.

Publication bias was checked using funnel plots looking at symmetrical distribution, and it was objectively verified using Egger’s regression test, which revealed that there was no publication bias (*p* < 0.98) (Appendix A).

This meta-analysis determined that the entomological outcomes of piperonyl butoxide long-lasting insecticidal nets effectively and efficiently reduced mean vector density per household per night by 3% compared to pyrethroid-only LLNs, demonstrating their effectiveness and efficacy over varying durations from one to three years post-distribution in Africa (Appendix A).

This study revealed that the pooled mean indoor vector density per household per night in the piperonyl butoxide long-lasting insecticidal nets intervention group was 1.9, which was significantly lower than the pyrethroid-only LLINs or control groups, with a mean of 8.04 per household per night, as shown in Appendix A.

Publication bias was checked using funnel plots looking at symmetrical distribution, and it was objectively verified using Egger’s regression test, which revealed that there was no publication bias (*p* < 0.148) (Appendix A).

This meta-analysis determined that the pooled mean indoor vector density per household per night for the pyrethroid-only long-lasting insecticidal nets or control group was 8.04 per household per night, as shown in Appendix A.

##### Post-Intervention Pooled Effectiveness and Efficacy of Pyriproxyfen Long-Lasting Insecticidal Nets (LLINs) Versus Pyrethroid-Only LLINs for Sporozoite Rate Reduction

Forest plots showed that pyriproxyfen long-lasting insecticidal nets (LLINs) post-intervention effectively and efficiently reduced the sporozoite rate by 15% compared to standard or pyrethroid-only LLINs (RR = −0.02 with a 95% CI of −0.08, 0.37), demonstrating their effectiveness and efficacy over varying durations from one to three years post-distribution in Africa. See details in Appendix A.

##### Pooled Prevalence of the Sporozoite Rate Among Pyriproxyfen LLINs Intervention

The pooled prevalence of the sporozoite rate was 165 per 100 mosquitoes (95% CI 1.13–2.18) in pyriproxyfen long-lasting insecticidal nets (LLINs) versus 227 per 100 mosquitoes in pyrethroid-only LLINs (95% CI 1.59–2.95) in post-intervention over varying durations from one to three years post-distribution in Africa (see details in Appendix A).

This meta-analysis determined entomological outcomes including effectiveness and efficacy in terms of the sporozoite rate reducing status, with the chlorfenapyr group being 9% effective with efficacy in reducing rates, as compared to the standard/control of the pyrethroid-only group, demonstrating their effectiveness and efficacy over varying durations from one to three years post-distribution in Africa. See details in Appendix A.

The pooled prevalence of the sporozoite rate was 79 per 100 mosquitoes (95% CI 0.49, 1.09) in chlorfenapyr long-lasting insecticidal nets (LLINs) versus 227 per 100 mosquitoes in pyrethroid-only LLINs (95% CI 1.59–2.95) in post-intervention over varying durations from one to three years post-distribution in Africa (see details in Appendix A).

This meta-analysis determined entomological outcomes including effectiveness and efficacy in terms of the sporozoite rate reducing status, with the piperonyl butoxide group being 10% effective and showing efficacy in reducing rates, as compared to the standard/control of the pyrethroid-only group in post-intervention over varying durations from one to three years post-distribution in Africa (see details in Appendix A).

The pooled prevalence of the sporozoite rate was 172 per 100 mosquitoes with 95% CI (1.06, 2.38) in piperonyl butoxide long-lasting insecticidal nets (LLINs) versus 227 per 100 mosquitoes in pyrethroid-only LLINs (95% CI 1.59–2.95) in post-intervention over varying durations from one to three years post-distribution in Africa (see details in Appendix A).

The pooled prevalence of the sporozoite rate was 227 per 100 mosquitoes in pyrethroid-only long-lasting insecticidal nets or control group (95% CI 1.59–2.95) in post-intervention over varying durations from one to three years post-distribution in Africa (see details in Appendix A).

##### Post-Intervention Entomological Inoculation Rate Among Different Long-Lasting Insecticidal Nets

This meta-analysis determined entomological outcomes including effectiveness and efficacy in the inoculation rate in terms of mean per household per night reduction, with the pyriproxyfen group showing a 7% reduction, as compared to the standard/control of the pyrethroid-only group in post-intervention over varying durations from one to three years post-distribution in Africa (see details in Appendix A).

This meta-analysis determined that the pooled mean entomological inoculation rate per household per night was 4 per 100 household per night with 95% CI (−0.00, 0.08) in the pyriproxyfen group compared to 7 per 100 household per night in the pyrethroid-only group with 95% CI 0.03,0.12) in post-intervention over varying durations from one to three years post-distribution in Africa (see details in Appendix A).

Publication bias was checked using funnel plots looking at symmetrical distribution, and it was objectively verified using Egger’s regression test, which revealed that there was no publication bias (*p* < 0.38) (Appendix A).

This meta-analysis determined entomological outcomes including effectiveness and efficacy in terms of the mean per household per night inoculation rate reduction reported in the chlorfenapyr group as reduced by 23%, as compared to the standard/control of the pyrethroid-only group in post-intervention over varying durations from one to three years post-distribution in Africa (see details in Appendix A).

This meta-analysis determined the pooled mean inoculation rate per household per night 4 per 100 household per night in chlorfenapyr LLNs intervention with 95% CI (−0.00, 0.08) versus 7 per 100 household per night (95% CI 0.03, 0.12) in post-intervention over varying durations from one to three years post-distribution in Africa (see details in Appendix A).

This meta-analysis determined entomological outcomes including effectiveness and efficacy in terms of the mean per household per night inoculation rate reduction, with the piperonyl butoxide group reduced by 12%, as compared to the standard/control of the pyrethroid-only group in post-intervention over varying durations from one to three years post-distribution in Africa (see details in Appendix A).

This meta-analysis determined the pooled mean inoculation rate per household per night 3 per 100 household per night with 95% CI (−0.00, 0.06) in piperonyl butoxide versus 7 per 100 household per night with 95% CI (0.03, 0.12) pyrethroid-only LLINs in post-intervention over varying durations from one to three years post-distribution in Africa (see details in Appendix A).

Publication bias was checked using funnel plots looking at symmetrical distribution, and it was objectively verified using Egger’s regression test, which revealed that there was no publication bias (*p* < 0.83) (Appendix A).

This review determined the pooled mean anopheles mosquito inoculation rate using pyrethroid-only group long-lasting insecticidal nets, which was 7 per 100 household per night with a 95% CI of −0.03, 0.12 in post-intervention over varying durations from one to three years post-distribution in Africa (see details in Appendix A).

We adopted GRADE (Grading of Recommendations Assessment, Development and Evaluation) as the method for assessing the quality of the body of evidence and for determining the direction and strength of the resulting recommendations. This generated evidence was evaluated in terms of the effectiveness and efficacy of pyriproxyfen, chlorfenapyr, and piperonyl butoxide long-lasting insecticidal nets against the pyrethroid-only LLINs (see details in Table 1).

The evidence generated from this meta-analysis reveals that pyriproxyfen (PPF) long-lasting insecticidal nets (LLINs) have no significant difference in malaria infection, case incidence, or anemia reduction among children, as compared to pyrethroid-only LLINs. However, this study found that pyriproxyfen (PPF) LLINs effectively and efficaciously reduced indoor vector density, entomological inoculation rate, and sporozoite rate of malaria parasites compared to pyrethroid-only LLINs.

This study found that chlorfenapyr (CFP) and piperonyl butoxide (PBO) long-lasting insecticidal nets (LLINs) are highly effective and efficacious in reducing malaria infection, case incidence, and anemia among children, as well as reducing indoor vector density, inoculation rate, and sporozoite rate in Africa as compared to pyrethroid-only LLINs.

Critical appraisal of individual randomized control trials revealed that 100% of the studies scored as having high quality, and Cochrane methodology was used to assess the risk of bias and evaluate evidence quality, which was graded as high. This research provides a very good indication of the likely effect. The likelihood that the effect will be substantially different is low (see details in Table 2).

The evidence generated from this meta-analysis reveals that pyriproxyfen (PPF) long-lasting insecticidal nets (LLINs) have no significant difference in malaria infection, case incidence, or anemia reduction among children as compared to pyrethroid-only LLINs. Based on the above finding, as an input, this study also evaluated the effectiveness and efficacy of chlorfenapyr and piperonyl butoxide long-lasting insecticidal nets compared to pyriproxyfen LLINs for malaria control in Africa.

##### Pooled Post-Intervention Effectiveness and Efficacy of Chlorfenapyr Long-Lasting Insecticidal Nets (LLINs) Versus Pyriproxyfen LLINs Malaria Infection Reduction in Africa

This study determined the overall pooled post-intervention effectiveness and efficacy of chlorfenapyr long-lasting insecticidal nets (LLINs) in malaria infection reduction in children, which was reduced by 1% as compared to pyriproxyfen LLINs (ARR = −0.01, 95% CI = −0.04, 0.03). See details in Appendix A.

The subgroup analysis of the post-intervention follow-up effectiveness and efficacy of chlorfenapyr long-lasting insecticidal nets (LLINs) showed a significant reduction in malaria infection among children over different durations, with a 2% reduction at 6 months and 1% at 36 months post-distribution of LLINs, compared to pyriproxyfen LLINs (ARR = −0.01, 95% CI = −0.04, 0.03). See details in Appendix A.

Publication bias was checked using funnel plots looking at symmetrical distribution, and it was objectively verified using Egger’s regression test, which revealed that there was no publication bias (*p* < 0.174) (Appendix A).

This study found no difference in the effectiveness and efficacy of piperonyl butoxide long-lasting insecticidal nets (LLINs) in reducing malaria infection risk in children over different durations compared to pyriproxyfen LLINs for malaria control (see Appendix A).

This meta-analysis determined entomological outcomes including effectiveness and efficacy of chlorfenapyr versus pyriproxyfen LLINs; the forest plot shows that chlorfenapyr long-lasting insecticidal nets significantly reduce pooled mean indoor vector density per household per night by 1% compared to pyriproxyfen LLINs in Africa. The included studies were homogeneous (*I*^2^ = 0.00%). See details in Appendix A.

Subgroup analysis with post-intervention follow-up effectiveness and efficacy of chlorfenapyr long-lasting insecticidal nets (LLINs) showed a significant reduction in pooled mean indoor vector density per household per night over different durations, with a 16% reduction at 12 months versus only 1% at 36 months post-distribution of LLINs, compared to pyriproxyfen LLINs (ARR = −0.01, 95% CI = −0.08, 0.06). See details in Appendix A.

Publication bias was checked using funnel plots looking at symmetrical distribution, and it was objectively verified using Egger’s regression test, which revealed that there was no publication bias (*p* < 0.272) (Appendix A).

This meta-analysis determined the entomological outcomes, effectiveness, and efficacy of piperonyl butoxide versus pyriproxyfen long-lasting insecticidal nets (LLINs). The forest plot shows that piperonyl butoxide long-lasting insecticidal nets significantly reduce indoor vector density per household per night by 4% compared to pyriproxyfen LLINs in Africa. The included studies were homogeneous (*I*^2^ = 0.00%). See details in Appendix A.

Subgroup analysis with post-intervention follow-up effectiveness and efficacy of piperonyl butoxide long-lasting insecticidal nets (LLINs) showed a significant reduction in pooled mean indoor vector density per household per night over different durations, with an 8% reduction at 12 months versus only 4% at 36 months post-distribution of LLINs, compared to pyriproxyfen LLINs (ARR = −0.01, 95% CI = −0.24, 0.16). See details in Appendix A.

Publication bias was checked using funnel plots looking at symmetrical distribution, and it was objectively verified using Egger’s regression test, which revealed that there was no publication bias (*p* < 0.974) (Appendix A).

This study reveals that chlorfenapyr LLINs effectively reduce the sporozoite rate by 7% compared to pyriproxyfen LLINs, demonstrating their superior efficacy in long-lasting insecticidal nets (LLINs), as shown in Appendix A.

The subgroup analysis based on a follow-up duration study analyzing the effectiveness and efficacy of chlorfenapyr long-lasting insecticidal nets (LLINs) showed a significant reduction in the pooled sporozoite rate per anopheles mosquito over different durations, with a 10% reduction at 12 months and 8% at 36 months post-distribution compared to pyriproxyfen LLINs. See details in Appendix A.

Publication bias was checked using funnel plots looking at symmetrical distribution, and it was objectively verified using Egger’s regression test, which revealed that there was no publication bias (*p* < 0.293) (Appendix A).

This study reveals that piperonyl butoxide long-lasting insecticidal nets (LLINs) effectively and efficaciously reduce the sporozoite rate by 1% compared to pyriproxyfen LLINs, as shown in Appendix A.

The subgroup analysis based on a follow-up duration study analyzing the effectiveness and efficacy of piperonyl butoxide long-lasting insecticidal nets (LLINs) showed a significant reduction in the pooled sporozoite rate per anopheles mosquito over different durations, with a 3% reduction at 12 months and 1% at 36 months post-distribution compared to pyriproxyfen LLINs. See details in Appendix A.

This study reveals that chlorfenapyr LLINs effectively reduce the mean entomological inoculation rate per household per night by 15% compared to pyriproxyfen LLINs, demonstrating their superior efficacy in mean entomological incubation rate reduction (see details in Appendix A).

The subgroup analysis based on a follow-up duration study analyzing the effectiveness and efficacy of chlorfenapyr long-lasting insecticidal nets (LLINs) showed a significant reduction in the pooled mean entomological inoculation rate per household per night over different durations, with a 13% reduction at 12 months and 33% at 36 months post-distribution compared to pyriproxyfen LLINs. See details in Appendix A.

Publication bias was checked using funnel plots looking at symmetrical distribution, and it was objectively verified using Egger’s regression test, which revealed that there was no publication bias (*p* < 0.38) (Appendix A).

The study reveals that piperonyl butoxide LLINs effectively reduce the mean entomological inoculation rate per household per night by 5% compared to pyriproxyfen LLINs, demonstrating their superior efficacy in mean entomological inoculation rate reduction (see details in Appendix A).

The subgroup analysis based on a follow-up duration study analyzing the effectiveness and efficacy of piperonyl butoxide long-lasting insecticidal nets (LLINs) showed a significant reduction in the entomological inoculation rate per household per night over different durations, with a 7% reduction at 12 months and 1% at 36 months post-distribution compared to pyriproxyfen LLINs. See details in Appendix A.

##### Malaria Incidence CFP vs. PPF

This study evaluated the post-intervention effectiveness and efficacy of chlorfenapyr long-lasting insecticidal nets (LLINs) in malaria case incidence reduction in children aged 6 months to 10 years over different durations, 1 year to 2 years, and overall reduction by 1%, as compared to pyriproxyfen LLINs (ARR = −0.01, 95% CI = −0.19, 0.17). See details in Appendix A.

The subgroup analysis based on a follow-up duration study analyzing the effectiveness and efficacy of chlorfenapyr long-lasting insecticidal nets (LLINs) showed a significant reduction in the entomological inoculation rate per household per night over different durations, with a 7% reduction at 12 months and 3% at 25 months post-distribution compared to pyriproxyfen LLINs. See details in Appendix A.

##### Malaria Incidence PBO vs. PPF

This study determined the post-intervention effectiveness and efficacy of piperonyl butoxide long-lasting insecticidal nets (LLINs) in malaria case incidence reduction in children aged 6 months to 10 years over different durations, 1 year to 2 years, and overall reduced by 2%, as compared to pyriproxyfen LLINs (ARR = −0.02, 95% CI = −0.57, 0.54). See details in Appendix A.

We adopted GRADE (Grading of Recommendations Assessment, Development and Evaluation) as the method for assessing the quality of the body of evidence and for determining the direction and strength of the resulting recommendations. This generated evidence was evaluated in terms of the effectiveness and efficacy of chlorfenapyr and piperonyl butoxide long-lasting insecticidal nets compared with pyriproxyfen long-lasting insecticidal nets for malaria control. The evidence generated showed that piperonyl butoxide (PBO) long-lasting insecticidal nets effectively and efficaciously reduce indoor vector density, entomological inoculation rate, and sporozoite rate of malaria parasites compared to pyriproxyfen (PPF) LLINs, but no significant difference was found in malaria infection reduction among children who use piperonyl butoxide (PBO) versus pyriproxyfen (PPF) long-lasting insecticidal nets in Africa.

This study found that chlorfenapyr (CFP) long-lasting insecticidal nets (LLINs) are highly effective and superiorly efficacious in reducing malaria infection, case incidence, and anemia among children, as well as reducing mean indoor vector density, mean entomological inoculation rate, and sporozoite rate compared to pyriproxyfen (PPF) long-lasting insecticidal nets (LLINs) in Africa.

Critical appraisal of individual randomized control trials revealed that 100% of the studies scored high quality, and Cochrane methodology was used to assess the risk of bias and evaluate evidence quality, which was graded as high. This research provides a very good indication of the likely effect. The likelihood that the effect will be substantially different is low (see details in Table 3).

## 5. Discussion

This systematic review and meta-analysis aims to inform governments, policymakers, health professionals, and populations at risk of malaria about the effectiveness and efficacy of long-lasting insecticidal nets (LLINs) and to evaluate changes in LLIN effectiveness and efficacy compared to pyrethroid-only LLINs. This study also compares the effectiveness of chlorfenapyr and piperonyl butoxide LLINs to pyriproxyfen LLINs for malaria control over time. The authors identified 11 RCTs that evaluated the effectiveness and efficacy of different long-lasting insecticidal nets (LLINs) as malaria prevention and control among children in Africa. This evidence synthesis adopted GRADE (Grading of Recommendations Assessment, Development and Evaluation) as the method for assessing the quality of the body of evidence and for determining the direction and strength of the resulting recommendations. The included studies compared the performance of pyriproxyfen, chlorfenapyr, and piperonyl butoxide long-lasting insecticidal nets against the pyrethroid-only LLINs.

This study revealed that chlorfenapyr long-lasting insecticidal nets (LLINs) among children in the treatment groups provided significantly better protection against malaria compared to the pyrethroid-only LLINs. On the other hand, the malaria infection risk reduction among children using the pyriproxyfen or piperonyl butoxide LLINs showed no significant difference compared to the pyrethroid-only LLINs.

More specifically, unlike pyriproxyfen or piperonyl butoxide, the chlorfenapyr long-lasting insecticidal nets (LLINs) provided significant advantages, including a reduction in malaria infection, the sporozoites rate and associated anemia compared to the standard pyrethroid-only group. The intervention resulted in protection against malaria by reducing the risk of malaria infection by 10%. In addition, the intervention reduced the sporozoite rate by 5% and anemia by 1%, as compared to standard or pyrethroid only. Moreover, this study determined that the post-intervention effectiveness and efficacy of chlorfenapyr long-lasting insecticidal nets (LLINs) in malaria infection risk reduction in children over different durations (6, 12, 18, 24, and 36 months) were reduced by 1%, as compared to the standard or control of the pyrethroid-only group.

Evidence shows that the prevention of and reduction in malaria transmission can be achieved by using vector control strategies, which include insecticide-treated mosquito nets [74]. Bed nets treated with an insecticide are believed to be much more protective than untreated nets [4]. However, we found that not all LLINs (pyriproxyfen or piperonyl butoxide LLINs) are effective. This finding is crucial as the vast majority of malaria cases and deaths are found in the WHO African Region, and the malaria impact continues to be a global challenge [75].

The reason why some of the LLINs appeared less effective than the standard can be explained in several ways. There is a reason to believe that resistant malaria species can infect children despite the bed nets [74]. Outdoor malaria transmission or poor utilization of the net could compromise the effectiveness and efficacy of the LLINs [76]. This implies that the distribution of these LLINs should be carefully considered, and further studies need to be carried out to uncover the main reasons.

As mosquitoes in many areas are now resistant to pyriproxyfen and piperonyl butoxide, nets treated with other active ingredients are needed to control malaria. On the other hand, the recent WHO recommendation encourages the use of new classes of dual-ingredient ITNs with different modes of action. Pyrethroid–chlorfenapyr nets combine a pyrethroid and a pyrrole insecticide to enhance the killing effect of the net, and pyrethroid–pyriproxyfen nets combine a pyrethroid with an insect growth regulator (IGR) that disrupts mosquito growth and reproduction [6]. Based on our study, the chlorfenapyr long-lasting insecticidal nets (LLINs) can still be considered.

Evolutionary theory reveals widespread resistance against insecticides, particularly pyrethroids, which were once used in bed nets to kill mosquitoes. This resistance was first detected in Côte d’Ivoire in 1993 and is now widespread throughout the region [74].

Some net products use piperonyl butoxide (PBO) in combination with pyrethroid insecticide to manage resistance, but the WHO does not consider these nets effective tools [4].

The WHO has released recommendations for two new insecticide-treated nets: pyrethroid–chlorfenapyr nets, which combine pyrethroid and chlorfenapyr insecticides to enhance net killing, and pyrethroid–pyriproxyfen nets, which use a pyrethroid and an insect growth regulator to disrupt mosquito growth and reproduction.

The World Health Organization (WHO) has issued a strong recommendation for the use of pyrethroid–chlorfenapyr ITNs over pyrethroid-only nets to prevent malaria in adults and children in areas with pyrethroid resistance. However, the WHO has also issued a conditional recommendation for the use of pyrethroid–chlorfenapyr ITNs instead of pyrethroid-PBO nets. Additionally, the WHO has issued a conditional recommendation against the use of pyrethroid–pyriproxyfen nets over pyrethroid-PBO nets [4]. Thus, our findings further justify and support the recommendations made by the WHO.

The evidence generated showed that piperonyl butoxide (PBO) long-lasting insecticidal nets effectively and efficaciously reduce indoor vector density, entomological inoculation rate, and sporozoite rate of malaria parasites compared to pyriproxyfen (PPF) LLINs, but no significant difference was found in malaria infection reduction among children who used piperonyl butoxide (PBO) versus pyriproxyfen (PPF) long-lasting insecticidal nets in Africa. These findings are strongly supported by a previous study, which revealed that in highly pyrethroid-resistant areas, unwashed pyrethroid-PBO nets led to higher mosquito mortality compared to unwashed standard-LLINs (risk ratio (RR) 1.84, 95% CI 1.60 to 2.11; 14,620 mosquitoes, five trials, nine comparisons; high-certainty evidence) and lower blood feeding success (RR 0.60, 95% CI 0.50 to 0.71; 14,000 mosquitoes, four trials, eight comparisons; high-certainty evidence) [77].

This study found that chlorfenapyr (CFP) long-lasting insecticidal nets (LLINs) are highly effective and superiorly efficacious in reducing malaria infection, case incidence, and anemia among children, as well as reducing mean indoor vector density, mean entomological inoculation rate, and sporozoite rate compared to pyriproxyfen (PPF) long-lasting insecticidal nets (LLINs) in Africa. This is supported by previous evidence that sleeping under a chlorfenapyr-PY LLIN offered superior protection against malaria infection [12]. The U.S. President’s Malaria Initiative, one of its five interventions, is to scale up access to and use of long-lasting insecticide-treated nets in 27 countries in sub-Saharan Africa [77]. Thus, this finding may inform the selection of types of LLITNs and the need to focus on monitoring.

The Sahel Malaria Elimination Initiative (SaME Initiative) aims to accelerate towards the attainment of malaria elimination goals by 2030 in the sub-region [78]. This could be realized when the latest evidence is considered, and this study could serve that and help planners to effectively base their plan on evidence, like the finding that both CFP and PBO effectively and efficaciously reduce malaria infection, case incidence, and anemia among children, indoor vector density, entomological incubation rate, and sporozoite rate reduction in Africa.

This review applied rigorous methodology, including GRADE and Cochrane tools, and focused on high-quality randomized controlled trials evaluating LLINs. Despite only 11 studies being included, their consistency allowed for meaningful synthesis. Key strengths include a focused scope and policy-relevant findings. Limitations include the small number of studies, all from Africa, limiting generalizability, some heterogeneity in outcome reporting, and a lack of long-term data on the resistance and sustainability of newer LLINs.

## 6. Conclusions

This systematic review and meta-analysis, guided by the GRADE framework and Cochrane risk of bias assessment, found high-quality evidence across all included randomized controlled trials. The consistency and strength of the findings suggest a low likelihood of substantial deviation from the observed effects.

This study found that pyrethroid-only LLINs (control arm) were associated with a higher pooled prevalence of malaria infection, case incidence, anemia, indoor vector density, entomological inoculation rate, and sporozoite rate compared to LLINs containing pyriproxyfen (PPF), chlorfenapyr (CFP), and piperonyl butoxide (PBO).

### 6.1. Significant Findings (Effectiveness and Efficacy)

Chlorfenapyr (CFP) LLINs demonstrated high effectiveness and superior efficacy in the following ways:Reducing malaria infection prevalence;Lowering malaria case incidence;Decreasing anemia in children;Reducing indoor vector density;Lowering entomological inoculation rate (EIR);Reducing sporozoite rate.Piperonyl butoxide (PBO) LLINs were also effective and efficacious in the following ways:Reducing malaria case incidence and anemia;Significantly lowering indoor vector density, EIR, and sporozoite rate;Outperforming pyriproxyfen (PPF) LLINs in entomological outcomes.Pyriproxyfen (PPF) LLINs showed significant entomological benefits, including the following:Reduced indoor vector density;Lower entomological inoculation rate (EIR);Decreased sporozoite rate;Non-significant findings (effectiveness and efficacy).PPF LLINs did not significantly reduce the following:Malaria infection prevalence;Malaria case incidence;Anemia in children, compared to pyrethroid-only LLINs.

When comparing PBO LLINs to PPF LLINs, no significant difference was found in malaria infection reduction, though PBO LLINs were more effective in reducing entomological indicators.

Chlorfenapyr (CFP) LLINs outperformed PPF LLINs across all key outcomes, confirming their superior efficacy in malaria control.

This evidence supports the prioritization of chlorfenapyr and piperonyl butoxide LLINs for malaria control in areas with pyrethroid resistance, while highlighting the limited epidemiological impact of pyriproxyfen LLINs, despite their entomological benefits.

### 6.2. Policy Recommendations

Policymakers and health planners should replace pyrethroid-only LLINs with chlorfenapyr or piperonyl butoxide LLINs, which demonstrated superior effectiveness and efficacy in reducing malaria burden. These nets should be prioritized in malaria-endemic regions, especially in areas with known pyrethroid resistance.

Net re-distribution cycles should be synchronized with the functional lifespan of LLINs to maintain optimal coverage and impact, even in settings where usage rates are low.

Malaria prevention strategies should emphasize protecting vulnerable groups, particularly children under 5 and pregnant women, by ensuring access to the most effective LLINs.

Resistance management should be integrated into national malaria control strategies. This includes rotating or combining LLINs with different active ingredients to preserve long-term efficacy.

### 6.3. Evidence-Based Decision Making

Policy decisions should be grounded in high-quality, context-specific evidence. The demonstrated superiority of CFP and PBO LLINs should guide procurement and distribution strategies.

### 6.4. Research Implications

The findings highlight the need for further research into the factors driving insecticide resistance and the net effect of resistance on LLIN performance.

Future studies should assess the real-world effectiveness of pyriproxyfen LLINs compared to CFP and PBO LLINs under diverse ecological and operational conditions to better understand their role in vector control.

## Figures and Tables

**Figure 1 ijerph-22-01045-f001:**
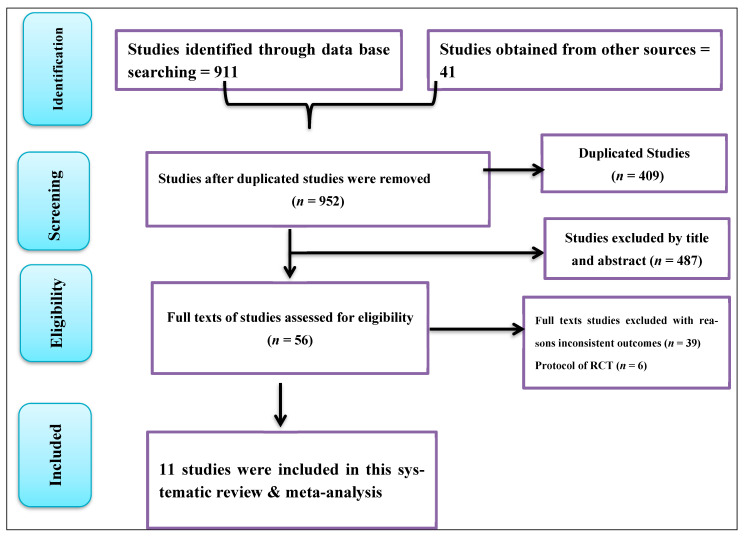
PRISMA flow chart showing the study selection process.

**Figure 2 ijerph-22-01045-f002:**
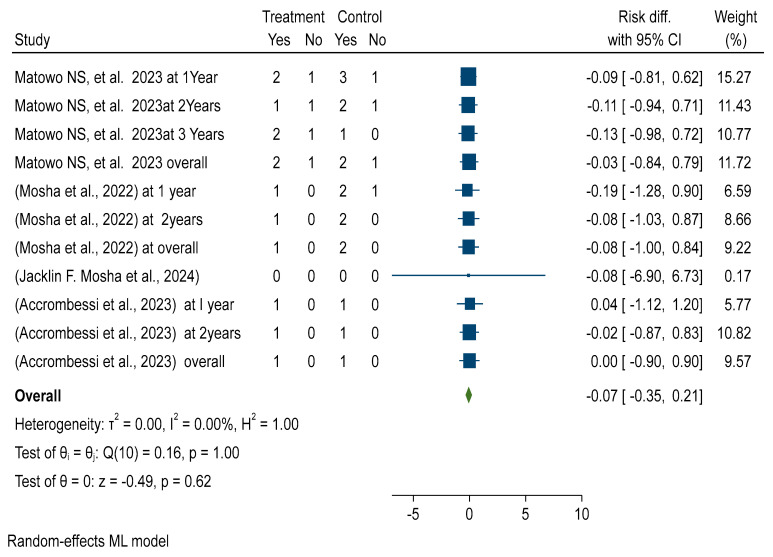
*Forest plot shows* pooled post-intervention effectiveness and efficacy of chlorfenapyr versus pyriproxyfen long-lasting insecticidal nets (LLINs), which reduce pooled sporozoite rate in Africa 2024 [8,10,16,28].

**Table 1 ijerph-22-01045-t001:** Evidence profile comparing the effectiveness and efficacy of pyriproxyfen, chlorfenapyr, and piperonyl butoxide LLINs with pyrethroid-only LLINs for malaria control in Africa.

**People**	All ages, adult or mixed (children and adults), included studies
**Settings**	Africa
**Intervention**	Effectiveness or efficacy of long-lasting insecticidal nets (LLINs) of pyriproxyfen, chlorfenapyr, and piperonyl butoxide
**Comparison**	Pyrethroid-only long-lasting insecticidal nets (LLINs)
Outcomes	Pyriproxyfen LLINs	Piperonyl butoxide LLINs	Chlorfenapyr LLINs	Pyrethroid-only long-lasting insecticidal nets (LLINs)	Certainty of the evidence (GRADE)
Malaria infection (9)	Pooled prevalence	Pooled prevalence	Pooled prevalence	40.84 Per 100 children(32.45%, 49.22%)	** 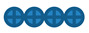 **
33.70 per 100 children(95% CI: 28.03–39.37%)	32.38 per 100 children(95% CI: 25.27–39.50%)	25.58 per 100 children(95% CI: 19.52–31.64%)	**High:**
Anemia (8),	Pooled prevalence	Pooled prevalence	Pooled prevalence	25.18 Per 100 children(12.78%, 37.58%)	** 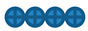 **
29.28 per 100 children(95% CI: 5.81–52.75%)	14.31 per 100 children(95% CI: 6.11%, 22.52%)	29.28 per 100 children(95% CI: 5.81–52.75%)	**High:**
Malaria case incidence per children years (4)	Pooled malaria case incidence	Pooled malaria case incidence	Pooled malaria case incidence	46 Per 100 children years(0.28, 0.63)	** 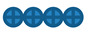 **
69 per 100 children years(95% CI: 0.46, 0.89)	31 per 100 children years(95% CI: 0.19, 0.43)	46 per 100 children years(95% CI: 0.28, 0.63)	**High:**
Mean indoor vectors/ vector density per household per night (8)	Pooled mean indoor vectors density	Pooled mean indoor vectors density	Pooled mean indoor vectors density	8.04 Per 100 Household(4.28%, 11.81%)	** 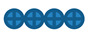 **
7.74 per 100 household(95% CI: 4.71, 10.78%)	1.9 per 100 household per night (95% CI: 1.15, 2.66%)	5.53 per 100 household per night (95% CI: 2.82, 8.15%)	**High:**
Mean entomological inoculation rate per household per night (6)	Pooled mean Inoculation Rate	Pooled mean MEIR	Pooled mean Inoculation rate	7 Per 100 Household(0.03, 0.12%)	** 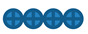 **
4 per 100 household (95% CI: (−0.00, 0.08%)	3 per 100 household per night(95% CI: 0.00, 0.06%)	4 per 100 household (95% CI: (−0.00, 0.08%)	**High:**
Sporozoite rate per mosquitoes (7).	Pooled sporozoite rate	Pooled sporozoite rate	Pooled sporozoite rate	227 Per 100 anopheles(1.59, 2.95%)	** 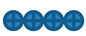 **
165 per 100 anopheles (95% CI: 1.13, 2.18%)	172 per 100 anopheles(95% CI: 1.06, 2.38%)	79 per 100 anopheles(95% CI: 0.49, 1.09%)	**High:**

Margin of error = Confidence interval (95% CI); RR: Risk ratio; GRADE: GRADE Working Group grades of evidence (see above and last page).

**Table 2 ijerph-22-01045-t002:** Evidence profile comparing the effectiveness and efficacy of pyriproxyfen, chlorfenapyr, and piperonyl butoxide LLINs with pyrethroid-only LLINs for malaria control in Africa.

**People**	All ages, adult or mixed (children and adults) included studies
**Settings**	Africa
**Intervention**	Effectiveness or efficacy of long-lasting insecticidal nets (LLINs) of pyriproxyfen, chlorfenapyr, and piperonyl butoxide
**Comparison**	Pyrethroid-only long-lasting insecticidal nets (LLINs)
**Outcomes**	Pyriproxyfen LLINsRelative effect (95% CI)	Piperonyl butoxide LLINsRelative effect (95% CI)	Chlorfenapyr LLINs Relative effect (95% CI)	Certainty of the evidence (GRADE)
Malaria infection (9)	0.0% no difference	1% less	−1% le S	** 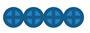 **
(−0.03,0.02%)	(−0.02, 0.01%)	(−0.04 to 0.02%)	**High:**
Anemia (8),	0.0% no difference	2% le S	1% le S	** 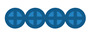 **
(−0.05 to 0.05%)	(−0.07 to 0.04%)	(−0.05 to 0.03%)	**High:**
Malaria case incidence per children years (4)	0.0% no difference	−3% le S	−4% le S	
(−0.11 to 0.12%)	(−0.57, 0.5%)	(−0.33, 0.26%)	**High:**
Mean indoor vectors/vector density per household per night (8)	−1% le S	−3% le S	−4% le S	** 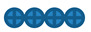 **
(−0.05, 0.08%)	(−0.19 to 0.13%)	(−0.15 to 0.06%)	**High:**
Mean entomological inoculation rate per household per night (6)	−7% le S	−12% le S	−23% le S	** 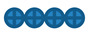 **
(−1.00 to 0.85%)	(−0.97, 0.73)	(−1.16 to 0.70%)	**High:**
Sporozoite rate per mosquitoes (7).	15% le S	10% le S	9% le S	** 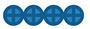 **
(−0.08, 0.37%)	(−0.09, 0.29%)	(−0.16, 0.35%)	**High:**

Margin of error = Confidence interval (95% CI); RR: Risk ratio; GRADE: GRADE Working Group grades of evidence (see above and last page).

**Table 3 ijerph-22-01045-t003:** Evidence profile comparing the effectiveness and efficacy of chlorfenapyr, and piperonyl butoxide long-lasting insecticidal nets with pyriproxyfen long-lasting insecticidal nets for malaria control in Africa.

**People**	All ages, adult or mixed (children and adults) included studies
**Settings**	Africa
**Intervention**	Effectivenessor efficacy of long-lasting insecticidal nets (LLINs) of chlorfenapyr, and piperonyl butoxide
**Comparison**	Pyriproxyfen long-lasting insecticidal nets (LLINs)
Outcomes	piperonyl butoxide LLINsRelative effect (95% CI)	Chlorfenapyr LLINs Relative effect (95% CI)	Certainty of the evidence (GRADE)
Malaria infection (9)	0.0% no difference	−1% le S	** 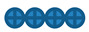 **
(−0.04,0.04%)	(−0.04 to 0.03%)	**High:**
Malaria case incidence per children years (4)	−2% le S	−1% le S	** 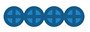 **
(−0.57, 0.54%)	(−0.19, 0.17%)	**High:**
Mean indoor vector density per household per night (8)	−4% le S	−1% le S	** 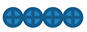 **
(−0.24 to 0.16%)	(−0.08 to 0.06%)	**High:**
Mean entomological inoculationrate per household per night (6)	−5% le S	−15% le S	** 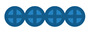 **
(−1.38, 1.48)	(−1.18, 0.88%)	**High:**
Sporozoite rate per mosquitoes (7).	−1 le S	−7% le S	** 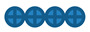 **
(−0.28, 0.26%)	(−0.35, 0.21%)	**High:**

Margin of error = Confidence interval (95% CI); RR: Risk ratio; GRADE: GRADE Working Group grades of evidence (see above and last page).

## Data Availability

All data generated or analyzed during this study are included in this article and its Appendix A.

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
