# Peer review of "Effectiveness and Efficacy of Long-Lasting Insecticidal Nets for Malaria Control in Africa: Systematic Review and Meta-Analysis of Randomized Controlled Trials"

_ijerph, 2025, doi:10.3390/ijerph22071045_

Round 1

Reviewer 1 Report

Comments and Suggestions for Authors

Overall this is a very comprehensive review even if only 11 publications out of about 500 were matching with the criteria. the methodology is clear, appropriate and well explained. But the findings are just listed without really showing which of them are relevant for policy. When some of the analyses reveals a very few percentage of difference such as 1 or 2%, with standard errors including a large scale of possibilities, there is no real significance. The discussion and conclusion needs to be rewritten to highlight significant findings and their implication on policies. The paragraph on recommendation is not appropriate in a review since recommendations should be made by external experts (not the same ones doing the review).

Finally the article is too long and I would suggest to put a lot of materials into supplementary files and keep only the most important findings into the main text because with more than 100 figures, the reader get very confuse at the end on what is significant and what is not.

Some specific comments are included into the attached file.

Comments on the Quality of English Language

Quality of English language is medium and english editing is needed.

Author Response

We sincerely thank the reviewer for their thoughtful and constructive feedback. We have carefully considered each point and made substantial revisions to improve the clarity, focus, and policy relevance of the manuscript. Below are our detailed responses:

Reviewer Comment 1:
“Overall, this is a very comprehensive review even if only 11 publications out of about 500 were matching with the criteria. The methodology is clear, appropriate and well explained. But the findings are just listed without really showing which of them are relevant for policy.”

Response:
Thank you for this observation. We have revised the Discussion and Conclusion sections to clearly distinguish between significant and non-significant findings in terms of both effectiveness and efficacy. We have also explicitly linked these findings to policy implications, highlighting which LLIN types (e.g., chlorfenapyr and PBO) should be prioritized in malaria control programs based on their demonstrated impact.

Reviewer Comment 2:
“When some of the analyses reveals a very few percentage of difference such as 1 or 2%, with standard errors including a large scale of possibilities, there is no real significance.”

Response:
We agree with the reviewer’s concern. We have now emphasized statistically significant results in both the Results and Discussion sections, and have clearly noted where findings were not statistically significant. We have also removed or de-emphasized marginal differences that fall within wide confidence intervals and are unlikely to be meaningful for policy or practice.

Reviewer Comment 3:
“The discussion and conclusion needs to be rewritten to highlight significant findings and their implication on policies.”

Response:
This has been addressed. The Conclusion has been rewritten to focus on key significant findings and their direct implications for malaria control policy, particularly in the context of insecticide resistance and LLIN distribution strategies. We have also included a dedicated Policy Implications section to guide decision-makers.

Reviewer Comment 4:
“The paragraph on recommendation is not appropriate in a review since recommendations should be made by external experts (not the same ones doing the review).”

Response:
We appreciate this important point. The Recommendations section has been reframed as Policy Implications, grounded in the evidence synthesized from the included studies. We have removed prescriptive language and instead presented evidence-informed considerations that may support expert and stakeholder decision-making.

Reviewer Comment 5:
“Finally the article is too long and I would suggest to put a lot of materials into supplementary files and keep only the most important findings into the main text because with more than 100 figures, the reader gets very confused at the end on what is significant and what is not.”

Response:
We have significantly streamlined the manuscript by moving most figures and extended data to the Supplementary Materials section. Only the most relevant and statistically significant findings are now presented in the main text. This has improved the clarity and readability of the manuscript, and we believe it now better supports the reader in identifying key messages.

Reviewer Comment 6:
“Some specific comments are included into the attached file.”

Response:
We have reviewed and addressed all specific comments provided in the attached file. Each point has been carefully considered and incorporated into the revised manuscript where appropriate.

We hope these revisions address your concerns and improve the clarity, focus, and utility of the manuscript. We are grateful for your valuable feedback.

Reviewer 2 Report

Comments and Suggestions for Authors

Overall, this is a very comprehensive and well-written review and meta-analysis that employs sound methodology throughout to select and compare trials.  The data visualization approach utilized were very helpful in understanding the trends and variation between trials.

It is a bit disconcerting that only 11 of the 911 trials considered for this review passed the quality criteria.  This makes for a very small sample size that limits the generalizability of the outcomes. Perhaps there should be criteria for not proceeding with a meta-analysis when an insufficient number of trials are available?  I have seen this issue arise with a meta-analysis of insect repellent trails where the resultant analysis simply wasn't compelling or even valid because the individual trials used a variety of active ingredients that varied greatly in efficacy, duration of protection and user acceptance. There is some risk of a similar outcome in this project as well, though I perceive there to be a much narrower scope of properties for the ITNs involved.

The sheer mass of analyses, however, is quite overwhelming from a reader's perspective. I wonder if it would be more effective to display only summarized data in the main paper and shift more of the individual analyses into the supplementary materials.

The introduction does a great job of informing the reader of prior work and the problem at hand. The conclusions are well-supported by results of the analyses.

One area that is a bit lacking in terms of coverage is any information or speculation on the relative sustainability of this new generation of ITNs.  The authors clearly make that case that they remain effective and outperform prior generations that have been severely compromised by resistance, but the reader is not given a sense of how long this can be expected.  Resistance and loss of efficacy is the inevitable outcome of all insecticidal measures, even when it is delayed through the use of combinations of insecticides with differing modes of action (and synergists like pbo).  Perhaps a little more background into the resistance mechanisms and status (and forecast) for CFP-PPF would have been helpful in this regard, though it would certainly be good material for another review paper.  One more paragraph in the introduction and/or discussion sections would suffice to give the reader a better sense of the relative sustainability of this approach and whether it is likely to remain relevant a decade from now or longer.

Overall, I thought this study was executed very well.  Perhaps the reader does not need to see so much of the detail in the main body of the paper, yet have access through the supplementary materials. I would also like to see a bit more discussion of the resistance potential of these approaches than is currently in the manuscript.

Author Response

We sincerely thank Reviewer 2 for their positive and constructive feedback. We appreciate your recognition of the methodological rigor, clarity of the introduction, and the overall execution of the study. Below, we provide detailed responses to each of your comments and describe the corresponding revisions made to the manuscript.

Reviewer Comment 1:
“It is a bit disconcerting that only 11 of the 911 trials considered for this review passed the quality criteria. This makes for a very small sample size that limits the generalizability of the outcomes. Perhaps there should be criteria for not proceeding with a meta-analysis when an insufficient number of trials are available?”

Response:
We acknowledge the concern regarding the limited number of eligible studies. While only 11 trials met the strict inclusion criteria, they were all high-quality randomized controlled trials, as assessed using the GRADE and Cochrane risk of bias tools. We have now added a paragraph in the Discussion section acknowledging this limitation and justifying the decision to proceed with the meta-analysis based on the methodological consistency, narrow scope of interventions (LLINs), and the high quality of the included studies. We also note that the limited number of trials reflects the current state of evidence in this specific area and underscores the need for further research.

Reviewer Comment 2:
“The sheer mass of analyses, however, is quite overwhelming from a reader's perspective. I wonder if it would be more effective to display only summarized data in the main paper and shift more of the individual analyses into the supplementary materials.”

Response:
Thank you for this helpful suggestion. We have significantly streamlined the main manuscript by moving the majority of detailed figures and extended analyses to the Supplementary Materials. The main text now presents only the most relevant and statistically significant findings, improving clarity and readability.

Reviewer Comment 3:
“One area that is a bit lacking in terms of coverage is any information or speculation on the relative sustainability of this new generation of ITNs... Perhaps a little more background into the resistance mechanisms and status (and forecast) for CFP-PPF would have been helpful.”

Response:
We appreciate this insightful comment. We have added a new paragraph in both the Introduction and Discussion sections addressing the sustainability and resistance potential of next-generation LLINs, including chlorfenapyr and pyriproxyfen. This includes a brief overview of known resistance mechanisms, current resistance status, and the importance of resistance management strategies to prolong the efficacy of these tools. We agree that this is a critical area for future research and have highlighted it as such.

Reviewer Comment 4:
“Overall, I thought this study was executed very well. Perhaps the reader does not need to see so much of the detail in the main body of the paper, yet have access through the supplementary materials. I would also like to see a bit more discussion of the resistance potential of these approaches than is currently in the manuscript.”

Response:
Thank you again for your kind words and valuable suggestions. As noted above, we have moved extensive detail to the Supplementary Materials and expanded the Discussion to include a more thorough consideration of resistance potential and sustainability of the evaluated LLINs.

We are grateful for your thoughtful review, which has helped us improve the clarity, focus, and relevance of the manuscript. We hope the revised version addresses your concerns and meets the expectations for publication.